# Innovative Pro-Smallholder Farmers' Permanent Mulch for Better Soil Quality and Food Security Under Conservation Agriculture

**Sibongiseni Mgolozeli** [1,2,*], **Adornis D. Nciizah** [1] , **Isaiah I. C. Wakindiki** [2] **and Fhatuwani N. Mudau** [2,3]

[1] Agricultural Research Council–Soil, Climate and Water, Private Bag X79, Pretoria 0083, South Africa; NciizahA@arc.agric.za

[2] Department of Agriculture and Animal Health, University of South Africa, Florida Park, Roodepoort 1709, South Africa; iwakindiki@gmail.com (I.I.C.W.); mudauf@ukzn.ac.za (F.N.M.)

[3] School of Agricultural, Earth and Environmental Sciences, University of KwaZulu-Natal, Private Bag X01, Scottsville 3209, Pietermaritzburg, South Africa

\* Correspondence: MgolozeliS@arc.agric.za; Tel.: +27-12-310-2653

**Abstract:** Soil degradation is the greatest threat to agricultural production globally. The practice of applying or retaining crop residues in the field as mulch is imperative to prevent soil erosion, maintain soil quality and improve crop productivity. However, smallholder farmers resort to maximizing profit by removing crop residues after harvest to sell or use them as feed for livestock. Agrimats are innovative pro-smallholder farming mulching materials that are manufactured using cheap or freely available organic waste materials. These materials include forestry waste, grasses, etc., therefore allowing smallholder farmers to make more profit through improved crop productivity for better food security. The most notable attributes of agrimats include their ability to prevent soil erosion, increase and sustain soil organic matter, suppress weeds, and conserve soil moisture. Food security challenge can be addressed by adopting agrimat technology as a sustainable permanent soil cover to improve soil quality and crop productivity. Agrimat incorporation in conservation agriculture practice could produce more food from less input resources (chemical fertilizers, water, etc.) with minimal or no adverse effect on the environment. This study aims to advocate permanent soil cover using agrimat as an innovative pro-smallholder farmer technology to improve soil quality for better food security.

**Keywords:** smallholder-farmers; agrimat; conservation agriculture; soil quality; food security

---

## 1. Introduction

The current global estimates indicate that about 80% of agricultural land suffers moderate to severe erosion [1]. Moreover, approximately 30% of the world's cropland has become unproductive and abandoned due to soil erosion [2], leading to one in every nine persons in the world being food-insecure [3] and about 66% of the global population being malnourished [1]. Ample evidence is available in literature on the perils of soil erosion as a significant threat to agricultural production more especially in semi-arid regions [4,5]. Sustainable land management strategies such as conservation agriculture (CA) have been identified and adopted in other parts of the world to prevent soil erosion, improve soil quality, conserve soil moisture and preserve the environment but most importantly, to match food production with the increasing world population [6–8]. A management system is considered to be CA if it consists of the following three principles; (i) permanent soil cover through crop

residues or mulch, (ii) minimum soil disturbance or no tillage, and (iii) crop diversification through crop rotation and/or cover crops [6,7].

Mulch is categorized as inorganic or organic with the latter being more commonly favored due to its biodegradable nature [5]. The inorganic types (gravel, polyethylene plastics, pebbles, etc.) of mulching material in agriculture are used mainly to control soil erosion and moderate soil moisture and temperatures, thereby increasing crop yield [9,10]. On the other hand, organic mulching materials (crop straw, grasses, sawdust, etc.) improve soil quality i.e., physical, chemical and biological characteristics by adding organic matter into the soil during the decomposition process [11]. The practice of applying or retaining crop residues in the field as mulch after harvest is imperative in maintaining soil health and productivity [7]. Permanent soil cover though mulching helps to reduce run-off by dissipating raindrop impact on the soil surface, thereby reducing soil erosion. Mango et al. [12] states that "Minimum soil disturbance and permanent soil cover help in improving soil organic matter content, reducing water run-off leading to increased infiltration, as well as increased biological activity". However, organic waste material applied as mulch in the field can be easily washed or carried away by adverse weather conditions (high wind speed and high-intensity rainfall) in semi-arid regions [13]. Thereby leaving the soil prone to (i) rain drop impact, which leads to soil erosion through runoff, (ii) high rates of evaporation due to high temperatures and (iii) poorly aerated soils because of low organic matter content in the soil. In addition, smallholder farmers practicing in sub-Saharan Africa (SSA) especially under semi-arid regions resort to maximizing profit by removing crop residues after harvest to sell or use them as feed for livestock, a practice that has led to poor adoption of CA in sub-Saharan Africa [14–16].

To address the problem of loose biological waste materials being easily washed away by adverse weather conditions, Onwona-Agyeman et al. [13] used a pressurized steam and compression technology to produce stable mulching materials called agrimats from forestry residues and placed them in gentle and steep slopes in the field. The results from the study by Onwona-Agyeman et al. [13] indicated that agrimats can reduce soil erosion by 94.4% and 92.3% on steep (30°) and gentle (5°) slopes respectively. Moreover, agrimats absorb and retain more moisture (67–77%) for up to two days. Agrimats are manufactured using cheap or freely available organic waste materials such as grass or weed biomass, municipal sewage sludge, algae residues, bagasse and forestry waste (thinned logs, woodchips, sawdust etc.) and therefore allow farmers to make more profit from selling crop residues after harvest as livestock feed or fuel [13].

This research study attempts to answer the following research questions: what is the best way to effectively use different and freely available organic materials as mulch to combat pervasive soil erosion, increase soil quality, improve crop productivity and food security with the minimum effects on the environment? What are the main challenges that prevent smallholder farmers from adopting conservation agriculture? Google scholar was used as a search engine to gather different scientific papers published over the past 20 years on the similar subject. Soil erosion, soil quality, permanent soil cover, agrimat, conservation agriculture, organic matter, crop productivity and food security were used as key words in the search engine to find relevant articles. The aim of this study is to assess the status of soil erosion and address it using CA as a conservation technique with more focus on permanent soil cover (agrimat), which improves soil quality for better, sustainable crop production, and food security.

## 2. Soil Degradation: The Biggest Threat in Agriculture

Soil degradation is a global challenge that threatens agricultural food production and food security [17]. According to Vaezi et al. [18], soil degradation is a consequence of the bare or scarcely covered soils in semi-arid lands. They further stated that "Approximately 40% of the world's land surface is classified as arid or semi-arid regions. About 35% of the lands in semi-arid areas are used for agricultural purposes." The decline in soil organic matter content due to drought and traditional agronomic practices employed by smallholder farmers has led to significant soil degradation in semi-arid parts of Southern Africa [18]. In South Africa alone, the changes in soil fertility status and

quality (loss of soil organic matter, declining N, soil acidity, expanding extent of saline and alkaline areas, soil acidity etc.) over the past three decades are worrying [19]. Moreover, South African soils are naturally very low in soil organic matter (SOM) content, with about 60% of the soils estimated to contain less than 0.05% SOM [20]. This could be the reason why over 70% of South Africa's land surface has been affected by varying intensities and types of soil erosion [21]. Although South Africa has various agro-ecological areas, it is generally classified as semi-arid region or water scare country since it receives less than 500 mm of rainfall per year on average [15]. Soil erosion through runoff and loss of soil moisture and sediments are the leading causes of soil degradation more especially in semi-arid regions [13,22]. According to Mohamadi and Kavian [23], soil erosion is an extremely dynamic and complicated process that is influenced by many factors, which include the topographic position of a slope, vegetation and soil type. Soil erosion occurs when raindrop kinetic energy overcomes the bonds holding soil particles together in the soil surface. The kinetic energy impact of raindrops on soil surface in high intensity storms causes increased soil particle detachment and sediment loss, excess runoff and surface sealing [23]. According to Khan et al. [24], the triggering of this process is related to slope, rainfall intensity and surface cover. The detached soil particles are transported away from the site of drop impact through runoff, which exacerbates land degradation [1,5]. Runoff is the principal erosive agent in water erosion processes that results in the loss of valuable plant nutrients together with soil sediments as well as soil water necessary for food production [2].

Figure 1, demonstrates the importance of leaving crop residues in the field under no-tillage systems in order to combat the devastating effect of soil erosion through runoff. The authors state that runoff rates were three to four times lower with a 50% reduction in sediment loss under a no-tillage system containing 50% residues compared to conventional tillage at the end of rainfall simulation events in the experiment shown in Figure 1. The soils (Regosols, Vertisols, and Calcisols) in the site where the experiment was conducted had poor physical and structural properties and were low in organic matter, which made them susceptible to both wind and water erosion. Moreover, the area is classified as an arid environment [22]. Figure 1 indicates that runoff is further exacerbated by increases in rainfall intensity. These results are in line with findings by Khan et al. [24], who reported an increase in sediment and water loss with an increase in rainfall intensity and slope steepness, in a laboratory experiment conducted under rainfall simulation using Calcaric Regosols. Xin et al. [25] also conducted a laboratory rainfall simulation experiment using black soils (Udic Argiboroll) at 7% fixed slope, five levels of residue cover (bare, 15%, 35%, 55%, and 75%) and four rainfall intensities (30 mm/h, 60 mm/h, 90 mm/h, and 120 mm/h). Their results indicated that residue cover strongly affects runoff, soil loss and infiltration. The authors reported that "The mean runoff reductions were 30.3%, 37.1%, 56.8%, and 72% for the 15%, 35%, 55%, and 75% residue covers compared to bare soil, respectively. The mean soil loss reductions were 41.5%, 58%, 89%, and 96% for the 15%, 35%, 55%, and 75% residue covers, respectively. When the rainfall intensity was higher, the protection effectiveness of the residues was weakened".

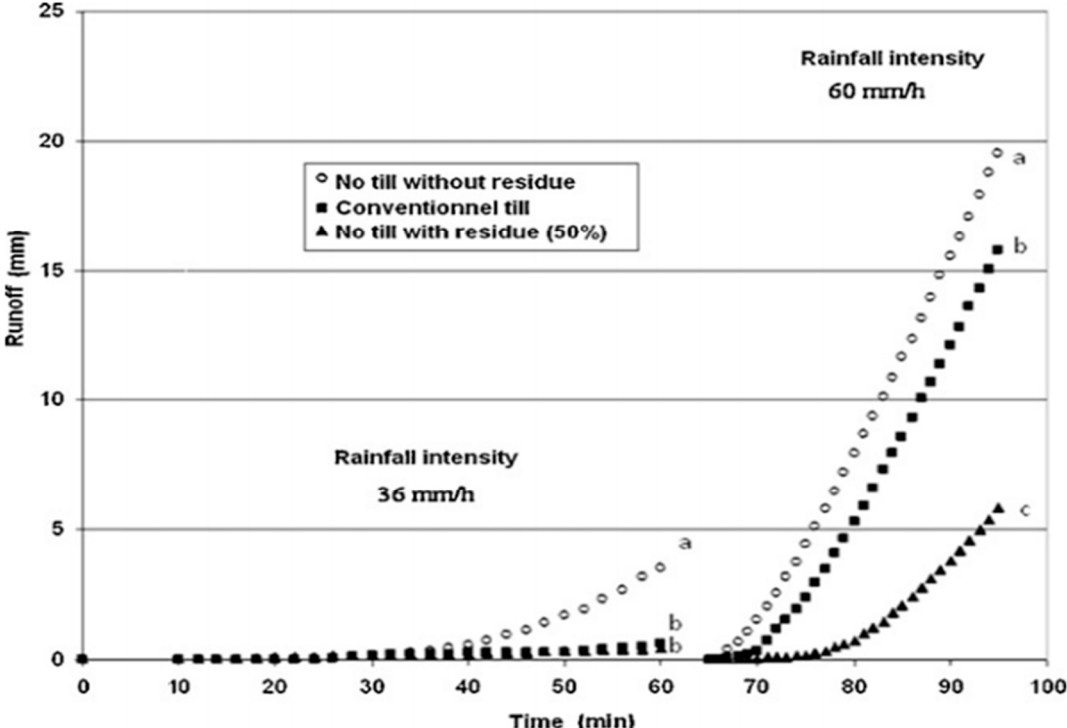

**Figure 1.** Runoff loss as affected by conventional tillage and no tillage with or without residue removal under two rainfall intensities in Zaers region [22].

Globally, erosion rates range from a low 0.001 t/ha/year on relatively flat land with grass or forest cover, to rates ranging from 1 to 5 t/ha/year in mountainous regions with natural vegetation. In croplands, however, global estimates for soil erosion ranges from 0.5 to 400 tons/ha/year and averages to approximately 30 tons/ha/year [1]. The numerous soil problems are further exacerbated by the seasonality and erratic distribution of rainfall, which results in varying periods of dry spells separated by wet periods [26]. Soil erosion and soil moisture loss are two major issues that need serious consideration when dealing with degraded soils especially in semi-arid regions as well as projects attempting to rehabilitate deserted or abandoned lands [13]. The loss of soil sediments (fertile top soil) together with runoff water is attributed to unstable soil structure due to drought and mostly, to erratic or erosive rainfall caused by climate change. Thierfelder et al. [18] states that "Conservation Agriculture (CA) is a key approach to address declining soil fertility and the adverse effects of climate change in southern Africa, however, CA alone is often not enough, and complementary practices and enablers are required to make CA systems more functional for smallholder farmers in the short and longer term."

## 3. Conservation Agriculture (CA) for Improved and Sustained Soil Quality, Crop Production and Food Security

Out of the global land mass of 13.2 billion ha, only 1.6 billion ha is currently in use for the cultivation of agricultural crops. Global population is projected to reach about 8.5 billion by 2030 and therefore an additional cropland of 81 to 147 million ha will be needed to meet global food demands [4]. Food and Agricultural organisation (FAO) estimates that by 2050, the demand of new croplands due to population pressure (of about 9.5 billion people), diet change and demand for biofuels is expected to reach approximately 3.2 billion ha, mostly at the expense of grasslands, forests and rangeland ecosystems. Shahid and Al-Shankiti. [27] further states that "By 2050, rising population and incomes are expected to result in a 70 percent increase in global demand for agricultural production". Gomiero [4] warns, "Concerning the future, we must take into account the potential effect of climate change on

soils and agriculture". On the other hand, it is also important to consider the impact that agriculture and soils have on climate change and global warming because current estimates indicate that 10–12% of global anthropogenic greenhouse gas emission are attributed to agriculture [28].

So how can the world produce enough food to feed 9.5 billion people in 2050 and achieve food and nutritional security with minimal or zero impact on climate change? Presenting during the Nobel Conference 54 held at Gustavus Adolphus College in Minnesota, USA, on the 2nd of October 2018, Rattan Lal stated that, inter alia, by "Increasing agronomic productivity from existing land, restoring degraded lands, enhancing biological nitrogen fixation by legumes and converting some agricultural land for nature conservancy without any conversion of natural land to agroecosystems, through eco-intensification and restoration of soil health". He defined eco-intensification as the strategy to produce more food from less land, per drop of water, per unit of input fertilizers and pesticides, per unit of energy, and per unit of carbon emission. Shahid and Al-Shankiti. [27] assert that, "Increased production is projected to come primarily from intensification on existing cultivated land, or on improving marginal lands, with irrigation playing a key role". The approach suggested by Rattan Lal can be achieved by employing proven and practical climate-smart agriculture techniques such as conservation agriculture, improved water and nutrient management, integrated nutrient management, improved grazing, intercropping livestock management, etc. [27]. The practice of increasing agronomic productivity from existing land without any conversion of natural land to agroecosystems is key to preserving the soil organic matter content. According to Giller et al. [29], conversion of land from forest or grassland to agriculture rapidly increases the rate of decline in soil organic matter (SOM), with up to 50% of the SOM being lost within 10 to 15 years.

In addressing crop productivity and food security while protecting the resource base with minimal effect on the environment, the formulation for the combination of conservation techniques must be site specific and should strictly be used as recommendations in other areas where similar soils and environmental/climatic conditions may exist. Conservation agriculture is one of the effective and sustainable land management strategies falling under climate-smart agriculture that can be adopted to achieve eco-intensification and restore soil quality. As stated by Malobane et al. [30], conservation agriculture is an agricultural management practice promoted in many regions worldwide because of its ability to enhance soil quality while conserving natural resources with minimal negative impact to the environment". Integrated nutrient management is another climate-smart conservation technique that combines the use of both organic and inorganic sources to rebuild soil organic matter in nutrient depleted soils, improve soil quality, increase crop yield and protect the natural resource base [27].

### 3.1. The Effect of CA on Soil Quality

Soil quality is the capacity of a soil to function within ecosystem boundaries, sustain biological productivity, maintain environmental quality, and promote plant and animal health [8]. The beneficial effects of CA in terms of better soil quality are reflected through an improvement in soil organic carbon in the top 10 cm soil depth [7,31], enhanced water infiltration rate as indicated in Figure 2 [22,32], enhanced water holding capacity [11,22], lower bulk density [11,33], higher aggregates stability [7,33] and better soil structure [33]. For improved water infiltration and capture, SOM can be maintained by limiting soil aggregates and structure breakdown through tillage practices. The results in Figure 2 were obtained in an experiment conducted by He et al. [32] in the semi-arid south central Shanxi province of China under Chromic Cambisol (sand 23.1%, silt 43.3% and clay 33.6%). The area receives mean annual temperature and rainfall of 10.7 °C and 555 mm, respectively. The researchers reported that total infiltration under no-till (NT) was greater, and the final (steady state) infiltration rate for NT plots (17.0 mm.min$^{-1}$) was four times that of the conventional tillage (CT) plots (4.25 mm.min$^{-1}$). Figure 2 indicates that no-till practice enables the soil to significantly absorb more water volumes since it has a better infiltration rate compare with the soil under CT practices. These results are congruent with the findings by Wang et al. [34] who investigated the effects of wheat stubble and traditional ploughing on runoff, infiltration, and soil loss in laboratory plots under rainfall simulation using clay loam soil in

Yanglin, China. The treatments in this experiment comprised of wheat stubble cover and traditional ploughing using 80 mm ha$^{-1}$ rainfall intensity for 1 h at three slope gradients (5°, 10°, and 15°). They found that the infiltration amount was higher under wheat stubble treatment (94.8–96.2%) than that from traditional ploughing (35.3–57.1%) and the trends were consistent at all three slopes. The practice of covering the soil surface using organic mulch suppresses runoff and increases the infiltration rate under various rainfall intensities more especially under no-till practice [6,25]. Conversion of conventional to conservation tillage, in line with the principles of CA, may improve soil structure, increase soil organic carbon, minimize soil erosion risk, conserves soil moisture, decrease fluctuations in soil temperature and enhance soil quality and its environmental regulatory capacity [6].

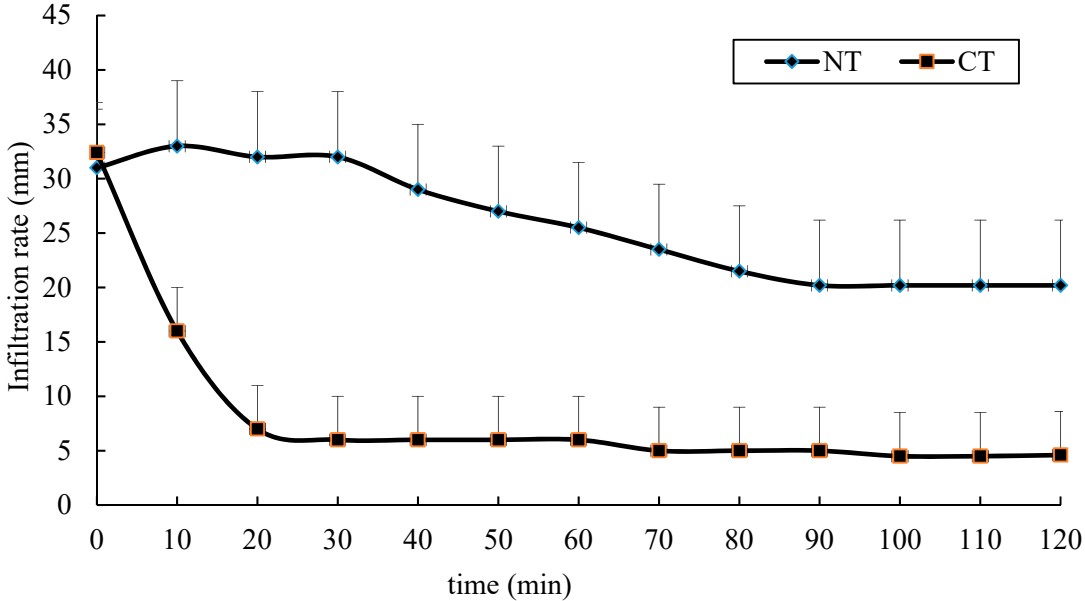

**Figure 2.** Changes in soil infiltration rate within 120 min under no tillage (NT) and conventional tillage (CT) treatments. LSD ($p < 0.05$). Redrawn from He et al. [32].

Figure 3, shows soil organic carbon content and bulk density in three different soil types (Vertisol, Cambisol and Luvisol) after 5 years under NT and CT treatments in a study conducted at the Merchouch Plateau in Morocco. The area is classified as a Mediterranean climate and receives a low average rainfall of 450 mm per year [31]. Contrary to many studies, the results shown in Figure 3 indicate that soil bulk density was significantly higher under CT in all three soil types after a period of five years compared to NT. This discrepancy can be explained using a statement from Castellini et al. [33] which says: "Several studies have found that implementation of NT results in over compaction of the soil but in other studies there was no significant soil compaction. In other words, when NT is used, soil compaction may still occur but this does not always cause a detrimental effect on crop production. In any case, its effect should always be assessed for a specific site, taking into consideration the type of agricultural cultivation and the soil types, as well as climatic condition". However, NT is an effective strategy in CA that can help to improve organic carbon (Figure 3) and carbon sequestration in the soil and ultimately reduce the negative impacts on climate [31,33]. In a study conducted by Dube et al. [7] on a Haplic Cambisol (62.4% sand, 16.0% silt, and 19.5% clay), it was demonstrated that in irrigated low-input systems as found in the Eastern Cape Province of South Africa, the levels of total SOM in the top 0–20 cm soil depth can be increased from as low as 10 g/kg to ranges above 20 g/kg after four years of CA. The site where this study was conducted is characterized by a warm temperate climate with average annual temperature and rainfall of 18.1 °C and 575 mm, respectively. SOM in the CA system is enhanced by the addition of mulch and/or the practice of leaving residues in the field as mulch, which, over time, decompose and increase the quality and overall fertility of the soil [7].

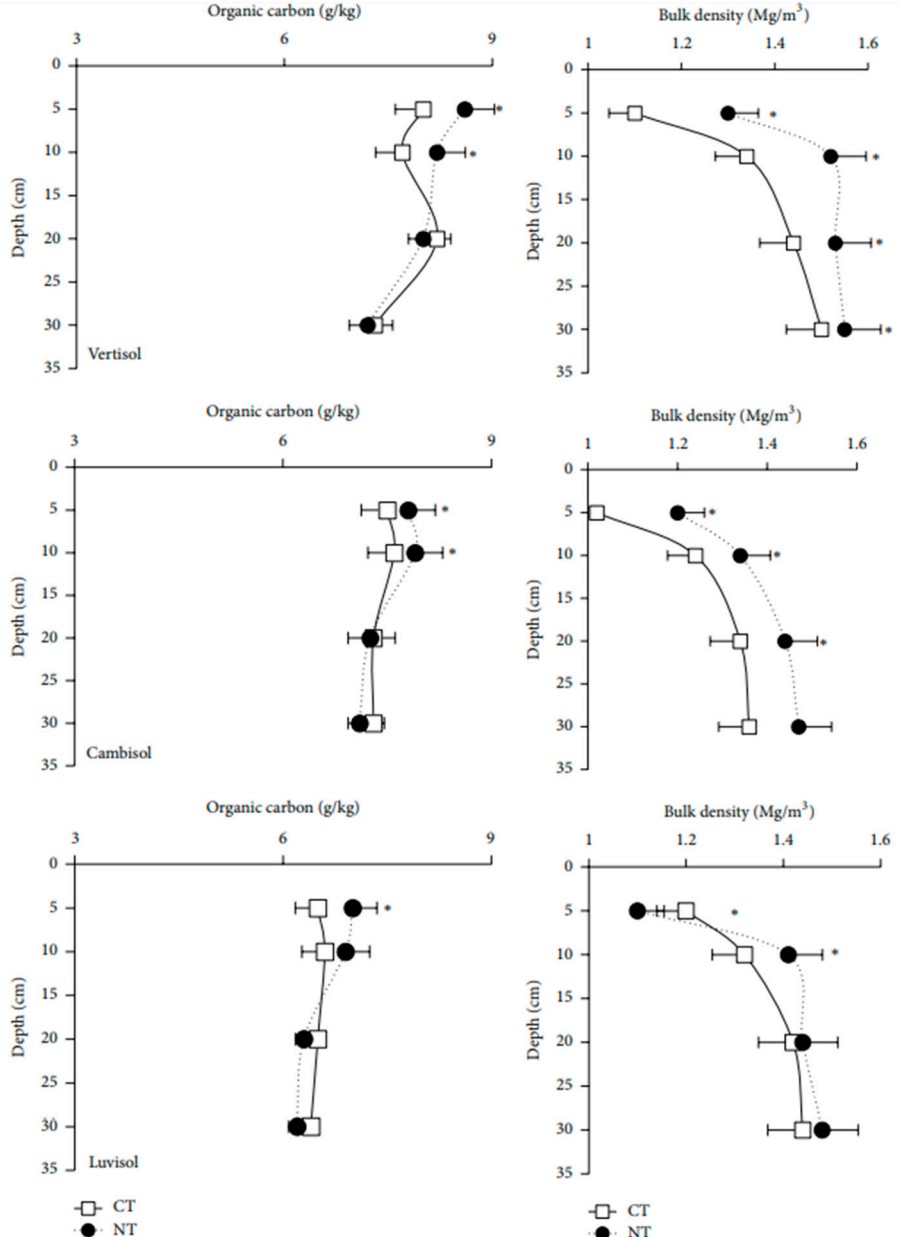

**Figure 3.** Soil organic carbon content (SOCc) and bulk density (Db) in three soil types after 5 years under NT and CT. At each depth (*) means the presence of significant differences between treatments ($p < 0.05$). The error bar represents one standard error [31].

In Rakai district, south central Uganda–an area dominated by Regosols (41.4%), Luvisols (22%) and Planosols (16.62%)–the decomposition of corn mulch that was applied at the thickness of 0.1 m by Kakaire et al. [35] resulted in 3.44%, 4.87% and 109.28% increase in field capacity (FC), permanent wilting point (PWP) and soil organic matter content, respectively. According to Kakaire et al. [35], in East Africa, especially in Uganda, Tanzania and Kenya, intercropping for green manure is rare because of the fear of competition for soil moisture. The disappearance of mulch on the soils has been worsened by human exports of crop residues, high termite activity and the long dry seasons. Reicosky. [36] state that "In all texture groups, as SOM content increased from 0.5 to 3%, available water capacity of the soil more than doubled. Increased water-holding capacity plus the increase in infiltration rate with higher SOM and decreased evaporation due to presence crop residues on the soil surface all contribute to

improve crop water-use efficiency". Soil water is a solvent utilized by plants to carry essential nutrients and a principal constituent of the growing plant, which also regulates soil temperatures [37].

Microorganisms also require water and favorable soil temperatures for metabolism and other activities such as decomposition of organic matter [38]. Soil temperatures, moisture and aeration are all essential for healthy soil and are the main edaphic factors controlling soil respiration [39]. In addition, water and nutrient (especially phosphorus) uptake is limited at temperatures below 10 °C [15]. In the wake of global warming, organic mulch plays a crucial role in soil productivity by moderating soil temperatures [9,40]. The combination of mulch and NT plays a crucial role in combating climate change by sequestering more carbon in the soil with minimal emissions of $CO_2$ into the atmosphere. As indicated in Figure 4, NT emits significantly lower $CO_2$ compared to other tillage systems. The study was conducted in an arid environment in Morocco, on a site dominated by soils of poor or no profile development (Rogosols), dark cracking clays (Vertisols), and calcareous soil (Calcisols). In addition, the area is characterized with low mean annual precipitation and high rates of evapotranspiration [22]. According to Paustian et al. [41], improved soil management can substantially reduce greenhouse gas emissions and sequester some of the $CO_2$ removed from the atmosphere by plants, as carbon (C) in soil organic matter. More than 25% of the greenhouse gas emissions attributed to agriculture could be reduced by employing soil management strategies such as zero or no-till [6] and the addition of plant derived C external sources like compost or biochar. Paustian et al. [41] noted that both compost and biochar are more slowly decomposed compared to fresh plant residue, with compost typically having a mean residence time several times greater than un-composted organic matter, and biochar mineralizes 10–100 times more slowly than uncharred biomass. Thus, a large fraction of added C–particularly for biochar–can be retained in the soil over several decades or longer, although residence time vary depending on the amendment type, nutrient content and soil conditions (such as moisture, temperature and texture). According to Davidson and Janssens. [42], the production of $CO_2$ in soils is almost entirely from root respiration and microbial decomposition of organic matter. Conservation Agriculture (CA) is part of climate smart agriculture that can be used to address climate change issues [41]. Currently, CA is being promoted in many regions of the world [30] especially in semi-arid areas, to rehabilitate degraded soils and improve ecosystem services [43]. The beneficial effect of CA reflects not only in terms of increased crop productivity and labor saving but it also helps in achieving environmental sustainability beside soil and land regeneration [11].

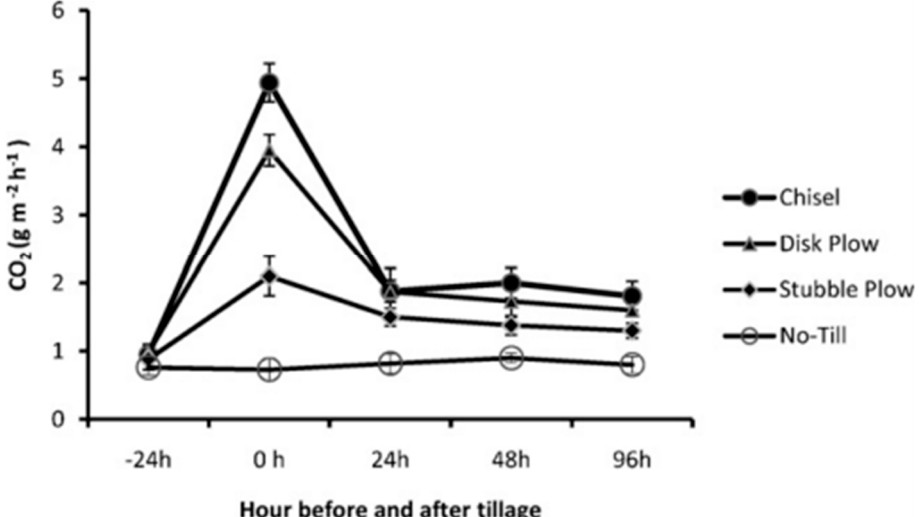

**Figure 4.** Soil $CO_2$ flux associated with primary full tillage as compared to no tillage (NT) systems [22].

### 3.2. The Role of CA in Crop Production and Food Security

The prevalence of undernourishment and food insecurity varies in different regions of the world. Currently, out of a world population of approximately 7.3 billion, about 66% of people are malnourished [1], with the majority of them located in sub-Saharan Africa and Asia [3]. In addition, The World Health Organization (WHO) and Food and Agricultural Organization (FAO) reports that one in every nine persons are food insecure. The mandate of FAO is to raise levels of nutrition, improve agricultural productivity to increase food security, better the lives of rural population and contribute to the growth of the world economy. Food security exists when all people at all times have physical, social and economic access to sufficient, safe and nutritious food to meet their dietary needs and food preferences for an active, healthy life [27].

Soil degradation especially through erosion has for a long time been a threat to food security since it reduces crop yields and causes soil abandonment in severe cases. The growing population further exacerbates soil degradation and this challenge needs to be addressed by focusing on increasing soil quality and soil sustainability so that the crop yields may be improved [11]. As stated by Yadav et al. [11], "Experts warn us that addressing the stagnating yields of our most important croplands is of paramount importance; failure to identify and alleviate the causes of yield stagnation, or reduction, will have a major impact on the future of global food security." Conservation agriculture (CA) is one of the climate-smart and sustainable land management strategies which helps to achieve eco-intensification and therefore increases global food security and nutrition [27]. The combination of CA principles can effectively improve soil quality; enhance food production and food security without any detrimental effects on the environment [30]. The goals of CA are to optimize crop productivity and farm income through maximum use of available resources and their effective recycling in the agroecosystem while arresting the adverse impacts on the environment [11]. In addition, CA is grounded in the principles of soil rejuvenation, envisioned to maximize the use efficiency of agricultural inputs e.g., seed, nutrient, water, energy, and labor, leading to higher profits to the grower [11]. However, in order for CA to work effectively in increasing crop production and hence reduce food insecurity, all the three principles of CA; 1, minimum soil disturbance; 2, permanent soil cover, and 3, crop rotation) must be followed first and performed properly [43].

The major challenge facing researchers and other proponents of CA is to convince smallholder farmers and other food producers to adopt CA as one of the effective techniques that helps to increase food security. Currently, the adoption rate of CA is very low, especially in Europe, Asia and Africa [7,11]. Out of the 1.6 billion ha currently in use for the cultivation of agricultural crops [18], CA occupies only approximately 156 million ha worldwide, increasing with the pace of 7 million ha annually mostly in the Americas [3]. Approximately 45% of the total area under CA is in South America, 32% in North America, 14% in Australia and New Zealand, 4% in Asia and 5% in the rest of the world including Europe and Africa. The top five pioneer countries leading the race in the adoption of CA are; United States of America (35 million ha), Brazil (31.8 million ha), Argentina (29 million ha), Canada (18.3 million ha) and Australia (17.6 million ha) [11]. In South Africa, the majority of farmers have not adopted CA yet, despite all its benefits. In the 2008/2009 planting season, only 7% of the total cultivated area was under no till [20]. Figure 5 show an estimate of the adoption status of CA in South Africa with Western Cape and Kwa-Zulu Natal provinces leading the race with more than 70% and 50–60% adoption rates, respectively. The areas with the lowest adoption rate of CA in the country include Springbokflats in the Limpopo province (10–20%), Eastern Cape province (approximately 5%), Easter Free State province (less than 5%), Orange River in the Northern Cape province (approximately 5%), and central Free State province (less 1%). Le Roux et al. [21] state that "at a scale of 1:2.5 million, Predicted Water Erosion Map (MWEP) indicates that a very large percentage of the Limpopo (60%) and Eastern Cape (56%) provinces are under severe threat to water erosion, whereas the Gauteng and North West provinces seem to be the least threatened by water erosion".

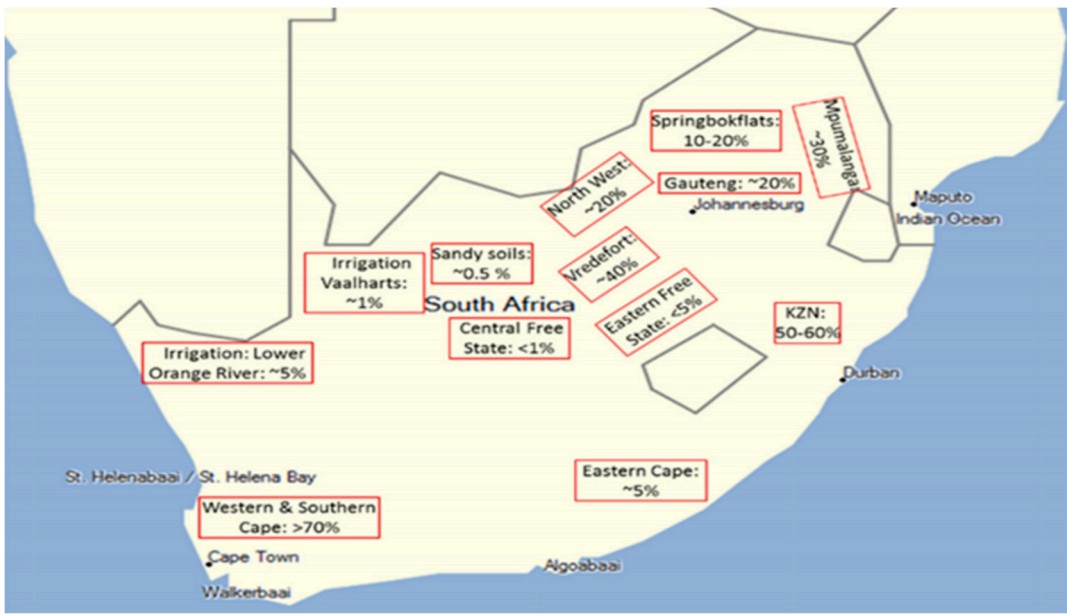

**Figure 5.** An estimate of CA adoption in South Africa [15].

Generally, the perils of soil erosion have led to soil loss from agroecosystems being 10–40 times faster than the processes of soil formation worldwide, exacerbating the problem of global food insecurity [1]. CA is among the top most effective and sustainable management strategies that prevent soil erosion, improve soil quality and conserve soil moisture but most importantly, to match food production with the increasing world population [7,8]. CA has the potential to significantly reduce soil erosion and improved land regeneration but all the three principles must be implemented in order for it to work effectively [11,43]. For instance, Giller et al. [14] state that "When no till is practiced in the absence of effective soil mulch cover, the effects can be disastrous with rapid surface sealing leading to increased run-off and accelerated soil erosion". On the other hand, researchers and other proponents of CA have failed to convince smallholder farmers and other poorly resourced food producers to adopt CA because they prioritize the use of crop residues for livestock feed rather than soil mulch cover and they prefer traditional tillage over no till for weed control purposes [14].

*3.3. Agrimats: The Innovative and Sustainable Mulching Materials*

Generally, mulch as a permanent soil cover decreases the effects of raindrop splashes on soil surfaces and hence slowing the detachment of soil particles and decrease the velocity of water on the soil surface [2,5]. Mrabet et al. [22] states that "Soil cover is the most important factor that influences water infiltration into the soil, thus reducing runoff and erosion." In addition, vegetation or canopy cover and plant residues protect the soil surface from water erosion by dissipating rainfall kinetic energy and therefore halt runoff and sediment loss [44]. The continuous accumulation of plant residues not only reduces runoff and erosion but also improves soil organic matter content, especially when adopted with minimum or no till and crop rotation as practiced in conservation agriculture [7,22]. Moreover, mulch conserve soil moisture by acting as a sediment trap that reduce surface runoff and enhance infiltration rate [5]. Dahiya et al. [45] conducted a field experiment to investigate the effect of straw mulch and tillage in soil water and temperature regimes on a silty loam Typic Hapludalf at the University of Hohenheim research center, Germany. They found that mulching decreased soil water loss on average by 0.39 mm per day, while rotary hoeing increased water loss on average by 0.12 mm per day compared with the control. In an experiment conducted by Alliaume et al. [46] in a Uruguayan sub-humid agro-ecological zone (with mean annual rainfall of 976 mm), in a with silty clay sub horizon and silty loam top soil, mulching increased water capture by 9.5% and reduced

runoff by 37%. In addition, reduction of runoff, under reduced tillage mulch when compared with conventional tillage was 33%, 39% and 27% during the tomato, sweet maize and onion crop growing seasons, respectively [46]. This indicates that mulching, no-till or reduced tillage practices or CA in general should not only be limited to dry (arid and semi-arid) regions because soil erosion threatens agriculture across different agro-ecological areas. According to Dahiya et al. [45], the major advantage of maintaining crop residues in sub-humid and semi-arid regions, is improved soil and surface conditions that allow better water infiltration.

The extreme weather events such as floods due to aggressive hailstorms with high-wind speed as a result of climate change are causing not only damage to the crops in semi-arid areas but they wash away all the loose organic materials used as mulch, together with top fertile soil through run-off [13,47]. This challenge has led to many agricultural soils in semi-arid areas becoming prone to (i) rain drop impact, which leads to soil erosion through runoff, (ii) high rates of evaporation due to high temperatures and (iii) poorly aerated soils because of low organic matter content in the soil [13]. The effect of soil erosion and impact of adverse weather conditions on mulching materials used as permanent soil cover could be reduced even by poor resourced smallholder farmers practicing in semi-arid regions. Onwona-Agyeman et al. [13] proposed and used a pressurized steam and compression technology to produce mulching materials called agrimats from different types of organic waste materials. In an experiment conducted at Gifu University in Japan to study soil erosion, Onwona-Agyeman et al. [13] placed agrimats on gentle and steep slopes in the field and their results indicated that agrimats could reduce soil erosion by 94.4% and 92.3% on steep (30°) and gentle (5°) slopes, respectively. Although the paper by Onwona-Agyeman et al. [13] does not describe the climatic conditions and soil properties of the study site, the area received 254.3 mm of rainfall during the four-month period of the experiment. Agrimats are designed to withstand high-speed runoff water and blowing away by heavy storms and strong winds respectively, as typically experienced in semi-arid environments. In addition, the agrimats serve as an innovative permanent and sustainable soil cover that lasts for at least two years before they completely decompose into the soil [13]. However, extensive research is needed, especially in Southern Africa; to critically compare and contrast the effectiveness of agrimats to other traditional soil amendments and mulching methods (especially compost, manures and biochars) across different climatic conditions and soil types. The current study puts emphasis on dry areas since they are more susceptible to crop failure and food insecurity due to global climate change [4]. The low adoption rate of conservation agriculture (CA) in semi-arid areas by smallholder farmers is attributed to the prioritization of crop residues for livestock use rather than soil mulching [14]. Theirfelder et al. [48] states that "Although the practice of CA can provide many benefits for smallholder farmers in Southern Africa, large-scale spontaneous adoption has been hampered by a number of constraints. The constraints include trade-offs between residue retention and livestock feed in mixed crop-livestock systems".

### 3.4. The Role of Agrimats in the Adoption of CA

Conservation Agriculture (CA) is defined as a method of managing ecosystems for improved and sustained productivity, increased profits and food security while preserving and enhancing the resource base and the environment [6]. Several researchers have proved over years that permanent soil cover is one of the most effective practices in CA systems that help to increase soil organic matter [6,7,31]. Since soil is a living entity, organic matter is to the soil what a heart is to the body–without organic matter, soil is dead. Increase in soil organic matter promotes aggregation, pore geometry and stability of aggregates as it acts as a binding/cementing agent and most importantly, it improves soil fertility and overall quality of the soil [7,49]. However, the major constraints keeping smallholder farmers from leaving and/or applying crop residues in the field as permanent soil cover is the prioritization of using crop residues to feed livestock instead of for mulching [14].

Agrimats could prove a viable option even for farmers who want to maximize profit since they are manufactured using freely available biomass instead of valuable crop residues that remain after grain

harvest. Agrimats of various sizes and thicknesses are manufactured using forestry waste (thinned logs, woodchips, sawdust etc.)  grass or weed biomass, municipal sewage sludge, algae residues, bagasse etc. [13]. Agrimats are laid out on top of the soil surface as organic mats or boards to serve as permanent soil cover that, inter alia, prevent soil erosion, preserve and increase soil moisture, moderate soil temperature regimes, improve soil quality and crop productivity as well as food security. However, it is important to note that other organic waste materials such as algae residues and municipal sewage sludge may contain high quantities of heavy metals that may pose hazards not only to crop growth but also to human health. Whilst preliminary work by Onwona-Agyeman et al. [13,50] has shown the advantages of agrimats over conventional mulch, extensive research is still needed to determine and quantify acceptable limits for the application of heavy metal containing organic materials in agricultural soils and how they can be incorporated into agrimat technology for better efficiency. In addition, the future research must also investigate the possibility of utilizing various weed materials classified as invasive species and/or allelopathic in different quantities together with environmentally friendly organic materials in the agrimat fabrication process. Future research will also need to look at the possibilities of reducing toxicity levels in heavy metals and the allelopathic effects of various organic materials using traditional methods such converting them into biochar or composting and innovative techniques prior to pressing them into agrimats. Moreover, future studies can investigate if the agrimats could prove a better and viable option for smallholder farmers as a soil conditioner compared to compost or traditional manures. Globally, there is dearth of information on how agrimats affect soil microbial population and activity, soil pH and other soil chemical properties including soil fertility in general, including in Japan where the agrimat mulching practice was first developed and tested. Onwona-Agyeman et al. [50] soaked urea-impregnated agrimats and compost-manufactured agrimats in water for 24 h and later found that urea-impregnated agrimats absorbed more water (77%) than the compost-manufactured agrimats (67%). Therefore, agrimats can also be used as a nitrogen use efficiency tool and provide a solution for semi-arid farmers weary of nitrogen fertilizers being lost through leaching as a result of erratic and erosive rainfall.

However, further field research is needed in the future to investigate the decomposition rates of agrimats made with various organic materials and how they influence soil biology (including soil microbial population and activity) and chemistry. Generally, the rate of decomposition and mineralization rate of any organic material is controlled by many factors, inter alia, soil moisture, soil microbial activity, temperature etc. [42,51]. Therefore, the utilization of agrimats may allow farmers to use their crop residues as livestock feed or sell them to other farmers for profit maximization. Several attempts by researchers to promote the adoption of CA to smallholder farmers in semi-arid regions has failed not only because of competition for crop residues, but also due to erratic and erosive hail storms that wash away loose organic mulching materials. In addition, the labor burden induced by no-tillage practice especially when herbicides are not used is unbearable for smallholder farmers. Rather, they believe that traditional tillage is a cost effective practice for weed control with minimal or no use of herbicides [14]. However, agrimats are an innovative technology that will eliminate not only the need to control weeds or use herbicides but also reduce the cost of irrigation in semi-arid regions where rainfall is scanty.

In addition, considering the physical status and durability of agrimats compared to loose organic residues, they can also be used to address the concerns pointed by Giller et al. [29], who stated that "retention of mulch is not always possible". They observed that in Mozambique, the mulch is often removed in a matter of weeks by termites. In such cases, agrimats mulching practice may delay the removal rate and allow the farmers more time to control termites effectively. Therefore, future research studies will look at the viability of injecting coated pesticides into the agrimats in order to minimize crop failure in areas with severe cases of termites and other pest/disease infestations. Although the agrimats serve as a permanent soil cover, they can be temporarily removed at planting to facilitate sowing and be laid back again on the soil surface after a week or so when the seedlings have fully developed. Currently, the most effective method is to place the agrimats between the rows

of growing seeding to cover the entire soil surface irrespective of inter-row plant spacing. This method does not only prevent water loss through evaporation from the soil surface but suppress weed growth below the agrimats as they act as a firm protective soil cover. Onwona-Agyeman et al. [13] states that "The use of agrimats as mulch could reduce weed growth from 0.5 tons/ha/month to 0.05 ton/ha/month". Moreover, "agrimats absorb and retain more moisture (about 70%) for up to 24 h". Despite all the benefits highlighted in this communication, one of the most important factors that is likely to influence the adoption of the agrimats is the economic viability of using agrimats versus prevailing practices. Extensive feasibility studies with the active participation of smallholder farmers still needs to be done to ascertain the costs of purchasing the agrimats and whether farmers are willing to take up this technology.

## 4. Conclusions

Soil erosion is a global phenomenon that negatively affects human health and environmental quality more especially in semi-arid regions. The combination of CA principles can effectively improve soil quality; enhance food production and food security with little or no detrimental effects on the environment and human health. No till is an effective strategy in CA that helps to maintain organic carbon and carbon sequestered in the soil and ultimately reduce the negative impacts of climate change. Most importantly, mulching is a practice that maintains and improves soil organic matter in the soil especially under CA systems. Although many farmers resort to maximizing profit by selling crop residues, or using them as livestock feed instead of leaving or applying them as mulch for soil organic matter improvements, the findings of this communication indicate that agrimat mulch could be employed by poor resource smallholder farmers as an alternative to their valuable crop residues. Agrimats are an innovative, sustainable and permanent soil cover mulching technology that promise to be a cost-effective option, which can be used to eliminate crop residue competition in the mixed crop-livestock systems since they are manufactured from freely available biological materials. In addition, agrimats eliminate, for smallholder farmers, the heavy burden of weed control and herbicide use when no tillage is employed and they reduce the cost of irrigation more especially in semi-arid areas where rainfall is scanty. However, more extensive research needs to be conducted to investigate several aspects of the agrimats for agricultural use and their efficiency over traditional mulching methods mostly in terms of cost effectiveness or economic viability, and to quantify the extent to which they can (i) improve soil quality (soil physical, chemical, and biological properties), (ii) enhance crop productivity (crop growth and biomass yields) in a sustainable manner and (iii) mitigate global food security especially for poor resource smallholder farmers in semi-arid or arid regions

In addressing crop productivity and food security while preserving the resource base with minimal effects on the environment, the conclusions and recommendations drawn from all future agrimat research work must be site-specific and should strictly be used as a guide for areas with similar soils and environmental/climatic conditions where the results will be obtained. Generally, global food security can be achieved by increasing agronomic productivity from existing land without any conversion of natural land to agroecosystems through eco-intensification and restoration of soil health. This includes the restoration of degraded lands through the incorporation of agrimats to CA systems, which employ biological nitrogen-fixing legumes in a rotation with other crops.

**Author Contributions:** S.M. gathered literature and wrote the original draft, A.D.N., I.I.C.W. and F.N.M. conceptualized the idea and revised the draft manuscript. All authors have read and agreed to the published version of the manuscript.

**Funding:** This research was funded by the Department of Science and Technology (DST), the Japan Science and Technology Agency (JST) and the Japan International Cooperation Agency (JICA).

**Acknowledgments:** The authors would like to thank DST and JST for funding this communication.

**Conflicts of Interest:** The authors declare no conflict of interest.

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
