# Peer review of "Innovative Pro-Smallholder Farmers’ Permanent Mulch for Better Soil Quality and Food Security Under Conservation Agriculture"

_agronomy, doi:10.3390/agronomy10040605_

Round 1

Reviewer 1 Report

The article concerns an important topic that is part of sustainable development. It concerns both soil protection, environmental protection, use of bio-waste, agricultural production and food safety. The structure of the paper is clear, but please pay attention to the correct signing of the figures (e.g. Figure 1 or Figure 1) and the use of the same signature throughout the text. In addition, I pay attention to the poor quality of the drawings presented at work. The literature review is rich, however, there is no information concerning the place of research when describing particular soil properties. This is so important because erosion and physicochemical properties are influenced by climatic conditions, among others. And the paper focuses on dry areas, therefore part of the research should take into account the results obtained under such conditions. This requires supplementing and providing information on where the individual data come from (research area).

In addition, attention should be paid whether the use of so-called agrimates from different bio-waste does not affect the lowering of soil pH, which is important, inter alia, for the availability of nutrients in the soil. Furthermore, when writing about smallholder farmers who do not leave crop residues in the fields, it should be considered whether it is economically profitable for them to use agrimates. Since the collection of crop residues is used on smallholder farmers to feed livestock, for example, farmers can afford to buy suitable organic mats. There is no doubt that education in this area is very important. Nevertheless, it should be noted that the topic is interesting and requires further detailed research.

Author Response

Reviewer number 1 Report:

Open Review

English language and style

( ) Extensive editing of English language and style required 
( ) Moderate English changes required 
(x) English language and style are fine/minor spell check required 
( ) I don't feel qualified to judge about the English language and style 

Yes

Can be improved

Must be improved

Not applicable

Does the introduction provide sufficient background and include all relevant references?

(x)

( )

( )

( )

Is the research design appropriate?

( )

(x)

( )

( )

Are the methods adequately described?

( )

(x)

( )

( )

Are the results clearly presented?

( )

(x)

( )

( )

Are the conclusions supported by the results?

( )

(x)

( )

( )

Comments and Suggestions for Authors

The article concerns an important topic that is part of sustainable development. It concerns both soil protection, environmental protection, use of bio-waste, agricultural production and food safety.

The structure of the paper is clear, but please pay attention to the correct signing of the figures (e.g. Figure 1 or Figure 1) and the use of the same signature throughout the text.

Response: Fig 1 and 2 were changed to Figure 2 and Figure 3 and are now consistent (same signature) with other Figures (Line 133 and 227).

In addition, I pay attention to the poor quality of the drawings presented at work.

Response: The quality of Figure 2 has been improved significantly by redrawing the whole scatter chat diagram (Line 227).

The literature review is rich, however, there is no information concerning the place of research when describing particular soil properties. This is so important because erosion and physicochemical properties are influenced by climatic conditions, among others. And the paper focuses on dry areas, therefore part of the research should take into account the results obtained under such conditions. This requires supplementing and providing information on where the individual data come from (research area).

Response: Additional information was added to describe the soil properties in various places, where different research studies were conducted, including climatic conditions and to further information was added to discuss the findings (See Line 117-131; Line 204–221; Line 242 – 247; Line 255 – 262; Line 276 – 294; Line 379 - 382; Line 384 -286; Line 411 - 413).

In addition, attention should be paid whether the use of so-called agrimates from different bio-waste does not affect the lowering of soil pH, which is important, inter alia, for the availability of nutrients in the soil.

Response: Further information was provided to indicate the current state of agrimat for agricultural use and possible future research areas in this regard have been highlighted. This include possible research experiments that will focus on how agrimats affects soil biology (and microbial population and activity) and chemistry such as pH including nutrient availability in the soil (Line 440 - 475).  

Furthermore, when writing about smallholder farmers who do not leave crop residues in the fields, it should be considered whether it is economically profitable for them to use agrimates. Since the collection of crop residues is used on smallholder farmers to feed livestock, for example, farmers can afford to buy suitable organic mats.

Response: The authors are cognizant of the economic viability of the agrimats for smallholder farmers and have indicated how they plan to accommodate these farmers in the pricing model of the agrimat in order to facilitate decision making that drive profit for the poor resource farmers (See Line 508-512). 

There is no doubt that education in this area is very important. Nevertheless, it should be noted that the topic is interesting and requires further detailed research.

Response: Many thanks to reviewer number 1, we added information on various sections of the manuscript mostly detailing research that still needs to be done on the agrimats (Line 86-89; Line 94-99; Line 107-108; Line 117-131; Line 178 – 182; Line 203 - 204; Line 341 – 351; Line 389 – 397; Line 508 - 524)

Response: We are not sure whether the following two comments were coming either coming from reviewer number 1 or number 2, but they were addressed accordingly.

Comment 1.

“Authors should discuss the results and how they can be interpreted in perspective of previous studies and of the working hypotheses. The findings and their implications should be discussed in the broadest context possible. Future research directions may also be highlighted” (Line 169-171 on the original manuscript from reviewers).

Response: We added detailed information, which further discusses the results using other studies as well future studies on the agrimats (Line 117-131; Line 203 - 225; Line 276 – 294; Line 389 – 397; Line 445 – 473).

Comment 2.

“In soils where nutrients are easily leached, chemical fertilizers such as urea could be injected into the agrimats during their fabrication process to i), improve moisture holding capacity, retard soil moisture loss and serve as a source of nitrogen for crops, thereby boosting their overall function as a permanent soil cover. Onwona-Agyeman et al. (2015) soaked urea-impregnated agrimats and compost manufactured agrimats in water for 24 hours and later found that urea-impregnated agrimats absorbed more water (77%) than the compost manufactured agrimats (67%). Therefore, agrimats can also be used as a nitrogen use efficiency tool and provide a solution for semi-arid farmers weary of nitrogen fertilizers being lost through leaching as a result of erratic rainfall.” Regarding the section above, either reviewer one or two stated that “This section is not mandatory, but may be added if there are patents resulting from the work reported in this manuscript.”

Response: Since there are not patents resulting from the manuscript, we decided to omit it (See Line 427-428).

Reviewer 2 Report

"Innovative pro-smallholder farmer’s permanent mulch for better soil quality and food security under conservation agriculture: A review" is an interesting paper describing the challenges of a nowadays agriculture, especially in the frame of a global soil erosion problem. A big emphasis is put on a Conservation Agriculture (CA) as a strategy to limit the organic matter loss from the topsoil horizon and to regain its pool in soil. 

The problem is thoroughly discussed based on the last two decades reports and reflects state of the art of scientific research. Nevertheless, I believe its scientific value can be increased  by adding more details on the proposed by the Authors agrimats and the way of their implementation: how are they introduced into the soil? Is it better than the popular forms of organic matter fertilizers e.g. manure, compost? Please also explain how the use of agrimats reduce weed growth (Line 353-356)? What is the legal status of the mats: organic fertilizer or soil improver? 

The manuscript is well-written and easy to follow. It needs a slight English correction, e.g:

L11: the greatest threat

L241: "been" doubled

L252: are to optimize

L386: fullstop missing

Author contributions: please fill it 

Taking into consideration all the above mentioned issues to be adressed I recommend to Accept the paper after minor revision.

Author Response

Reviewer number 2 Report:

Open Review

English language and style

( ) Extensive editing of English language and style required 
( ) Moderate English changes required 
(x) English language and style are fine/minor spell check required 
( ) I don't feel qualified to judge about the English language and style 

Yes

Can be improved

Must be improved

Not applicable

Does the introduction provide sufficient background and include all relevant references?

(x)

( )

( )

( )

Is the research design appropriate?

(x)

( )

( )

( )

Are the methods adequately described?

(x)

( )

( )

( )

Are the results clearly presented?

(x)

( )

( )

( )

Are the conclusions supported by the results?

( )

(x)

( )

( )

Comments and Suggestions for Authors

"Innovative pro-smallholder farmer’s permanent mulch for better soil quality and food security under conservation agriculture: A review" is an interesting paper describing the challenges of a nowadays agriculture, especially in the frame of a global soil erosion problem. A big emphasis is put on a Conservation Agriculture (CA) as a strategy to limit the organic matter loss from the topsoil horizon and to regain its pool in soil. 

The problem is thoroughly discussed based on the last two decades reports and reflects state of the art of scientific research.

Nevertheless, I believe its scientific value can be increased by adding more details on the proposed by the Authors agrimats and the way of their implementation

Response: More detailed information on the scientific values of the agrimats was added on Lines 440 - 475 and Lines 485 - 497

 how are they introduced into the soil?

Response: Agrimats are laid out on top of the soil surface as organic mats or boards to serve as permanent soil cover that, inter alia, prevent soil erosion, preserve and increase soil moisture, suppress weed growth etc (Line 440 - 443; Line 448 – 497; Line 508 - 524)

Is it better than the popular forms of organic matter fertilizers e.g. manure, compost?

Response: Initial work by Onwona-Agyeman showed the benefits of using agrimats over conventional mulch, however, we included future research that still needs to be done on the agrimats (Line 450 - 455)

Please also explain how the use of agrimats reduce weed growth (Line 353-356)?

Response: Although the agrimats serve as a permanent soil cover, they can be temporality removed at planting to facilitate sowing and be laid again on the soil surface after a week or so when the seedlings have developed. Currently, the most effective method is to place the agrimats between the rows of growing seeding to cover the surface irrespective of inter-row plant spacing (Line 495 - 497).

What is the legal status of the mats: organic fertilizer or soil improver? 

Response: The mats serve both as a soil improver/protector in a sense that they improve soil organic matter, soil fertility as decompose over time therefore increase the overall soil quality (physical, chemical and biological properties (Line 440-475)

The manuscript is well-written and easy to follow. It needs a slight English correction, e.g:

L11: the greatest threat (

Response: …’a greatest threat’ was changed to “the greatest threat’: Line 11)

L241: "been" doubled (One “been” was eliminated in the sentence: Line 312)

L252: are to optimize (are to optimizing was changed to “are to optimize: Line 323)

L386: full stop missing (full stop was inserted: Line 397)

Author contributions: please fill it (Authors contribution was filled: Line 533 - 535)

Taking into consideration all the above mentioned issues to be addressed I recommend to Accept the paper after minor revision.

Response: We thank you for your kind comments, we hope the manner in which we addressed the comments is satisfactory

Response: We are not sure whether the following two comments were coming either coming from reviewer number 1 or number 2, but they were addressed accordingly.

Comment 1.

“Authors should discuss the results and how they can be interpreted in perspective of previous studies and of the working hypotheses. The findings and their implications should be discussed in the broadest context possible. Future research directions may also be highlighted” (Line 169-171 on the original manuscript from reviewers).

Response: We added detailed information, which further discusses the results using other studies as well future studies on the agrimats (Line 117-131; Line 203 - 225; Line 276 – 294; Line 389 – 397; Line 445 – 473).

Comment 2.

“In soils where nutrients are easily leached, chemical fertilizers such as urea could be injected into the agrimats during their fabrication process to i), improve moisture holding capacity, retard soil moisture loss and serve as a source of nitrogen for crops, thereby boosting their overall function as a permanent soil cover. Onwona-Agyeman et al. (2015) soaked urea-impregnated agrimats and compost manufactured agrimats in water for 24 hours and later found that urea-impregnated agrimats absorbed more water (77%) than the compost manufactured agrimats (67%). Therefore, agrimats can also be used as a nitrogen use efficiency tool and provide a solution for semi-arid farmers weary of nitrogen fertilizers being lost through leaching as a result of erratic rainfall.” Regarding the section above, either reviewer one or two stated that “This section is not mandatory, but may be added if there are patents resulting from the work reported in this manuscript.”

Response: Since there are not patents resulting from the manuscript, we decided to omit it (See Line 427-428).